# The Expression of TP63 as a Biomarker of Early Recurrence in Resected Esophageal Squamous Cell Carcinoma after Neoadjuvant Chemoradiotherapy

**DOI:** 10.3390/biomedicines12051101

**Published:** 2024-05-16

**Authors:** Chih-Hung Lin, Po-Liang Cheng, Cheng-Yeh Chuang, Yu-Ting Kang, Li-Wen Lee, Tzu-Hung Hsiao, Chung-Ping Hsu

**Affiliations:** 1Division of Thoracic Surgery, Department of Surgery, Taichung Veterans General Hospital, Taichung 40705, Taiwan; sphenoidlin@gmail.com (C.-H.L.); chuang5045@vghtc.gov.tw (C.-Y.C.); 2Department of Pathology and Laboratory Medicine, Perelman School of Medicine, University of Pennsylvania, Philadelphia, PA 19104, USA; po-liang.cheng@pennmedicine.upenn.edu; 3Department of Medical Research, Taichung Veterans General Hospital, Taichung 40705, Taiwan; polove908@gmail.com (Y.-T.K.); liwenjoyce1112@gmail.com (L.-W.L.); 4Research Center for Biomedical Science and Engineering, National Tsing Hua University, Hsinchu 30013, Taiwan; 5Department of Public Health, Fu Jen Catholic University, New Taipei City 242062, Taiwan; 6Institute of Genomics and Bioinformatics, National Chung Hsing University, Taichung 40227, Taiwan; 7Division of Thoracic Surgery, Department of Surgery, Buddhist Tzu Chi General Hospital, Hualien 97004, Taiwan

**Keywords:** esophageal squamous cell carcinoma, neoadjuvant concurrent chemoradiotherapy, RNA sequence, TP63, recurrence

## Abstract

Esophageal cancer ranks among the ten most common cancers worldwide. Despite the adoption of neoadjuvant concurrent chemoradiotherapy (nCCRT) followed by surgery as the standard treatment approach in recent years, the local recurrence rate remains high. In this study, we employed RNA-seq to investigate distinctive gene expression profiles in esophageal squamous cell carcinoma (ESCC) with or without recurrence following a standard treatment course. Our findings indicate that recurrent ESCC exhibits heightened keratinizing and epidermis development activity compared to non-recurrent ESCC. We identified TP63 as a potential candidate for distinguishing clinical outcomes. Furthermore, immunohistochemistry confirmed the trend of TP63 overexpression in ESCC recurrence. Patients with elevated TP63 expression had poorer overall survival and lower 3-year recurrence-free survival. This study underscores the potential of TP63 as a biomarker for detecting cancer recurrence and suggests its role in guiding future treatment options.

## 1. Introduction

Esophageal cancer ranks amongst the ten most common cancers in the world, with high incidence in developing countries. It is one of the most aggressive tumors and its growth is relatively rapid. The main histological subtype of esophageal cancer is esophageal adenocarcinoma (EAC) in Western areas [1,2], whereas 90% of esophageal cancer is esophageal squamous cell carcinoma (ESCC) in Taiwan. ESCC is mainly located in the upper to middle esophagus and is derived from the epidermis. Its risk factors are highly correlated with unfavorable habits, including a low intake of fruits and vegetables, smoking, excessive alcohol consumption, and betel nut chewing [3,4].

While several treatments, such as surgical resection of esophageal tumor, chemotherapy, and radiation therapy, have been applied to improve survival rate in patients with esophageal cancer, the efficacy of a single treatment remains poor [5]. Currently, multimodality treatment is predominantly employed, and with this approach, surgical resection remains the most effective treatment choice for patients with resectable esophageal cancer [6]. However, the most common cause of treatment failure in ESCC patients after surgery is recurrence [7]. Over the past few decades, neoadjuvant concurrent chemoradiotherapy (nCCRT) followed by surgery has been proven to prolong the survival of patients with resectable advanced esophageal cancer [8,9] and become the standard therapy in advanced esophageal cancer [10]. Nevertheless, the recurrence rate still ranges from 40 to 60%, and the prognosis of patients with recurrence is poor [11,12]. Although some clinical and pathological information may be related to tumor recurrence, the prediction is still unsatisfactory. Therefore, seeking early prognosis biomarkers associated with early esophageal cancer recurrence is important and instrumental to help decision making for effective treatment strategies.

A growing amount of research indicates that several genes such as *CLDN4* [13], *RUNX3* [14] and *E-cadherin* [15] could be potential ESCC prognostic biomarkers. Furthermore, *CLDN4* [13] and *AGR2* [16] could be a CCRT response indicator. However, none of them could be used to detect the potential early recurrence of ESCC after nCCRT followed by surgery. Genomic studies show that *TP63*, *SOX2*, and *KLF5* consist of the core regulatory circuit that controls epigenetic and transcription patterns in ESCC [17]. TP63 is a transcription factor from the p53 family, and the TP63 mutation is much more commonly detected and has elevated expression in ESCC [18]. It also regulates the growth of esophageal squamous carcinoma cells through many other pathways, such as the β-catenin/c-Myc signaling pathway and the AKT signaling pathway [19,20]. However, few studies have focused on the relationship between TP63 and the ESCC early recurrence.

In this study, we leverage high-throughput next-generation sequencing to identify predictable prognostic biomarkers of early recurrence in esophageal squamous cell carcinoma. We demonstrate that the expression of TP63 is significantly correlated with early tumor recurrence in ESCC.

## 2. Materials and Methods

### 2.1. Patients and Specimen Collection

In this study, 15 patients with esophageal squamous cell carcinoma who underwent surgical resection after neoadjuvant CCRT were enrolled. They were classified based on their recurrence status during the follow-up period into non-recurrence and recurrence groups. Criteria of no recurrence included the following: (1) tumor resolution on CT; (2) grossly no tumor on endoscopic examination, and biopsy at the residual scar tissue or original tumor sites yielded a negative result for malignancy; (3) no documented distant metastases by PET-CT examinations. Frozen specimens were obtained from the BioBank of Taichung Veterans General Hospital (TCVGH) and utilized for RNA sequencing. Additionally, immunohistochemistry was employed to validate the tissue expression of the target gene. Tissues from an additional 50 patients were collected from the TCVGH Biobank, and clinicopathological information, including age, sex, histological subtype, and treatment response, was obtained from delinked medical records. All samples used in this study from March 2008 to January 2021 were approved by the TCVGH Institutional Review Board. The IRB number for RNA-seq is CE17279A and the IRB number for IHC is CF21046A.

### 2.2. RNA Sequencing and Gene Expression Analysis

The total RNA was extracted from frozen esophageal squamous cell carcinoma tissues using the RNeasy Mini Kit (Qiagen, Venlo, Limburg, The Netherlands), following the manufacturer’s instructions. RNA libraries were prepared using the TruSeq Stranded mRNA library Prep kit for cDNA reverse transcription and library construction (Illumina, SanDiego, CA, USA). Subsequently, the libraries were sequenced using HiSeq 2500 sequencers (Illumina, San Diego, CA, USA). To analyze the sequencing data, sequence reads were aligned to the human reference genome GRCh38 using the HISAT2 aligner tool (version 2.2.1) [21]. Read counts were calculated using featureCounts (version 2.0.1) [22], and differential gene expression profiles identified using the R package DESeq2 (R version 4.0.2; DEseq2 version no. 1.40.2) [23]. A gene was considered significant if its log2 fold change was >1 or <1, and the DESeq-adjusted *p*-value was <0.05. A principal component analysis (PCA) was also performed by the function “plotPCA” in R package DESeq2. The first two PCs were used to plot the results.

For gene set enrichment analysis (GSEA), the *p*-value metrics of all genes were converted to log form, and the positive or negative sign was added based on the fold change estimate obtained from DESeq2. The genes were then pre-ranked by the transformed value before being input into GSEA software (version 3.0) (PMID: 16199517, PMID: 12808457). Gene sets of the gene ontology biological process (GO:BP) with sizes above 15 and under 500 were used in the analysis.

### 2.3. Immunohistochemistry

To detect the expression of p63 using IHC analysis, ESCC tissues were formalin-fixed and paraffin-embedded. Following tissue sectioning, deparaffinization and rehydration were achieved using xylene and different concentrations of ethanol. Antigen retrieval was performed with citrate-hydrochloric acid. Non-specific protein blocking involved the use of 10% goat serum. The anti-p63 antibody (ab735, Abcam, Cambridge, UK) was then applied at a 1:50 dilution, followed by treatment with an HRP-conjugated secondary antibody. Subsequently, diaminobenzidine was used for coloration, and counterstaining was conducted with hematoxylin. Two experienced pathologists from VGHTC independently evaluated the staining intensity of TP63 in tumor cells. Staining intensity is scored on a scale of 0–2, with 0 indicating no expression, 1 indicating mid expression, and 2 indicating strong expression.

### 2.4. Statistical Analysis

The two-tailed Student’s *t*-test was used for analyzing the differences between two continuous variables and chi-square test for categorical variables. A *p*-value < 0.05 was considered to be significant. All data analyses were conducted using R (version 4.0.2).

## 3. Results

### 3.1. Characteristics of the Study Population

Fifteen patients were divided into two groups according to the recurrence status of ESCC. Eight (53.33%) patients experienced a recurrence within 47 to 182 days (mean, 141.5 ± 43.5 days) and seven patients did not experience a recurrence. All of these patients were male, and the average age of the patients was 56.60 ± 6.68 years. The pathologic stage was classified according to the tumor–node–metastasis (TNM) classification (*TNM Classification of Malignant Tumours*, 8th edition) [24]. Most recurrence developed in stage III and IV. The average age (*p* = 0.84, *t*-test) and the cancer stage distribution (*p* = 0.2, chi-squared) between recurrent and non-recurrent groups were similar (Table 1).

### 3.2. RNA Expression Analysis of ESCC Recurrence Genes

We performed RNA sequencing analysis to identify the possible prognosis biomarkers in ESCC-recurrence patients. The results obtained for the principal component analysis (PCA) are described in Figure 1A. The PC1 and PC2 values form two separate clusters between patients with and without a recurrence of ESCC; PC1 captures 50% of the overall variance and PC2 accounts for 31%. To evaluate the transcriptomic difference between the non-recurrence and recurrence groups, we identified 1132 up-regulated and 162 down-regulated differentially expressed genes (DEGs) using DESeq2 (Figure 1B). The top 20 up-regulated differential expression genes are presented in Table 2.

### 3.3. Keratinocyte Proliferation Was More Activated in Non-Recurrent ESCC

We utilized gene set enrichment analysis (GSEA) to explore the difference in biological process between ESCC with or without recurrence. The top 10 enriched up-regulated gene sets were associated with processes such as keratinization, proliferation, and epidermis development, as visually depicted in Figure 2. This implies a higher level of abnormal cell proliferation and differentiation in recurrent ESCC compared to non-recurrent ESCC. It is noteworthy that, despite the identification of some down-regulated genes in the comparison between non-recurrent and recurrent ESCC, no enriched down-regulated gene sets were discerned, emphasizing the specificity of the observed molecular signatures.

### 3.4. High TP63 Expression Is Related to Early ESCC Recurrence and Poor Prognosis

From the top 20 differentially expressed genes and the activated keratinization proliferation and epidermis development biological processes in recurrent ESCC, we identified TP63 as a putative biomarker to predict the clinical outcome. Through immunohistochemical (IHC) staining, we evaluated TP63 expression in 50 ESCC tissues, half of which had recurrent (25 patients) and the other half non-recurrent (25 patients) clinical outcomes. The clinical pathological characteristics of these 50 ESCC patients are shown in Appendix A. These 25 patients with recurrence experienced recurrence within 25 to 235 days (average: 135.2 ± 53.5 days). All patients had an average age of 54.52 ± 7.71 years, including 48 males and 2 females.

We classified the IHC staining results into three levels: strong expression (>80%), mid expression (<80%), and no expression (Figure 3). The tissue sections were all judged by professional pathologists, and the parts stained with TP63 antibody were all tumor cells. The distribution of the TP63 expression levels in recurrent and non-recurrent ESCC was significantly different (Table 3, *p* value < 0.01; chi-square test). Following surgery, the survival and recurrence status of patients were continuously monitored, and survival curves were plotted. In comparison to the group combining no expression and mid expression of TP63, patients with strong TP63 expression exhibited a poorer prognosis (*p* = 0.21) (Figure 3G). Furthermore, the 3-year recurrence-free survival rate in the strong-TP63-expression group was notably lower than that in the low-expression group (*p* = 0.22) (Figure 3H). Summarizing the above results, we confirm that TP63 is more commonly expressed in recurrent ESCC patients, and those with high TP63 expression have a lower postoperative survival rate.

## 4. Discussion

In this study, we explored the differential gene expression between ESCC patients with or without recurrence after neoadjuvant CCRT followed by surgical resection. We found significant gene expression differences between the two clinical outcomes, which enriched some biological processes such as the abnormal growth and development of the epidermis in ESCC-recurrent patients. Furthermore, we demonstrated that TP63 protein expression tends to appear in patients who will suffer subsequent ESCC recurrence.

In clinical practice, the recurrence of esophageal cancer remains challenging to avoid and is often disheartening. A clinical trial examined 170 patients with stage I and stage II esophageal cancer. In this study, even with early-stage esophageal cancer receiving CCRT followed by surgery, the recurrence rate was still 31% [25]. There was already evidence suggesting that certain factors, such as tumor differentiation, resection margin, nodal status, and the response to neoadjuvant CCRT, may be associated with the recurrence of esophageal cancer [26,27]. However, the prediction of esophageal cancer recurrence remains inadequate. The use of biomarkers may provide a resolution. The discovery of the relationship between tumor recurrence and TP63 not only provides us with more information, but through further research into its correlations, it may help identify additional mechanisms contributing to tumor recurrence. Combining it with clinical data may potentially enhance the accuracy of predicting tumor recurrence.

We found several top up-regulated genes with high gene expression levels, such as KRT1, KRT17, ALDH3A1, S100A7A, and TP63 in recurrent ESCC (Table 2). Previous studies showed that oral squamous cell carcinoma with low ALDH3A1 expression was significantly worse than its high-ALDH3A1-expression counterpart [28], and its overexpression could drive cancer stem cell expansion and impair immune surveillance [29]. Furthermore, ALDH3A1 was identified as a novel downstream target of ESCC core regulatory circuitry [17]. On the other hand, keratins are used as diagnostic biomarkers in multiple cancers. A recent study found that a higher expression of KRT1 and KRT17 is associated with low overall survival in melanoma [30]. KRT17 could also activate AKT signaling and induce EMT in ESCC [31]. An abnormal expression of S100A7 has been observed in various cancers, including up-regulation in the tissues and blood of ESCC patients. Through the action of the S100A7/JAB1 axis, it facilitates the migration and proliferation of ESCC cells. Additionally, S100A7 has been found to promote the infiltration of M2 macrophages and the process of angiogenesis [32]. TP63 has a double role, functioning in either an oncogene or a tumor suppressor gene in different situations [33]. It could regulate tumor cell proliferation via the AKT signaling pathway and stimulate tumor cell invasion and metastasis in ESCC [19,20]. Therefore, we identified several differentially expressed genes associated with poor prognosis in tumors from ESCC patients who experienced recurrence after CCRT. These distinct genes may represent crucial factors contributing to the recurrence, highlighting their potential significance in understanding the underlying mechanisms of recurrent ESCC.

The cell fate of the epidermis depends on the balance of inputs from pro-survival and pro-differentiation cascades [34]. Therefore, aberrant epidermis development leads to epidermal cancer development. In this study, we found recurrent ESCC had excessive keratinization and epidermis development activity compared to non-recurrent ESCC. However, other studies indicate that there are different clinical outcomes that appear in skin cancers. For example, there were no significant differences in postoperative recurrence between keratinization and non-keratinization subtypes in lung squamous cell carcinoma [35]. Furthermore, more recurrences were discovered in patients with oral squamous cell carcinoma with low keratinization [36]. As previous studies have shown, TP63 is crucial to the development of the epidermis [37] and essential for the epithelial stem cell’s proliferative potential [38]. Therefore, the abnormal TP63 expression might cause aberrant epidermis development. In our findings, the abnormal TP63 expression tended to be discovered in recurrent ESCC after neoadjuvant CCRT treatment, which implies the TP63 has the potential to promote ESCC recurrence. However, the mechanism that causes the different responses to aberrant keratinization between these epidermal-derived cancers still needs more studies.

A cancer that is deemed recurrent usually reappears after a period in which the cancer cannot be detected. Therefore, a cancer stem cell may hide and harbor in the microenvironment until recurrence happens. TP63 expression is regulated through multiple complex pathways, such as Notch, Hedgehog, WNT, FGFR2, and EGFR [39]. The elevated expression of TP63 in recurrent ESCC corresponds to the final output of misregulation of these pathways. Previous research also found that TP63 cooperated with other transcription factors such as NF-κB to influence the tumor microenvironment (TME) [40]. The microenvironment might be changed under the regulation mechanism of ESCC and cause possible chemoresistance and subsequent cancer recurrence.

There are still several limitations in our study. First, it was a small-scale retrospective case–control study. The use of a case–control design may have led to selection bias, especially with the limited number of samples. This study aimed to confirm the correlation between recurrence and biomarkers, and we were concerned that a cohort study approach might result in too few recurrence cases to effectively address our research question. Therefore, we chose the case–control method for numerical matching. In the future, we also hope to further validate our findings by expanding the sample size, evolving into a prospective study, or utilizing multi-center data to enhance the representativeness and generalizability of our results. Second, our study only enrolled ESCC; further research is needed to confirm whether this applies to EAC. Third, with the development and widespread adoption of novel therapies, such as immunotherapy, it remains necessary to gather more data to confirm their applicability in the future.

In summary, our study revealed the gene expression difference between different clinical outcomes of ESCC. Most importantly, we identified TP63 as a biomarker for predicting early recurrence in ESCC. Therefore, our findings could provide a new aspect for therapeutic strategy making in the future.

## 5. Conclusions

This study explores the differences in gene expression between groups of patients with ESCC who received neoadjuvant concurrent chemoradiotherapy combined with surgery. Our findings show that the TP63 gene and protein expression levels are significantly higher in the recurrence group, accompanied by increased keratinizing and epidermis development activity. A higher expression of TP63 is associated with lower overall survival rates and a three-year recurrence-free survival. We hope that TP63 can serve as a biomarker for predicting early recurrence in ESCC and provide a new direction for future treatment strategies (Figure 4).

## Figures and Tables

**Figure 1 biomedicines-12-01101-f001:**
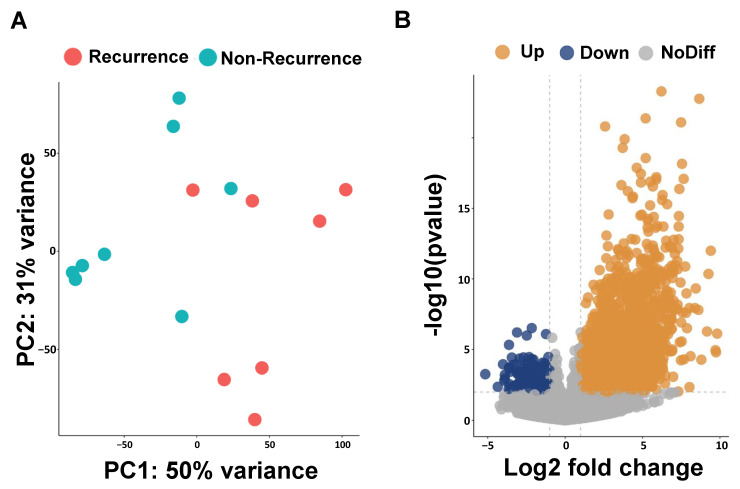
Principal component analysis (PCA) and volcano plots of differentially expressed genes (DEGs). (**A**) PCA plot reveals two clusters of esophageal cancer patients with and without recurrence. (**B**) Volcano plot shows DEGs of esophageal cancer patients with and without recurrence.

**Figure 2 biomedicines-12-01101-f002:**
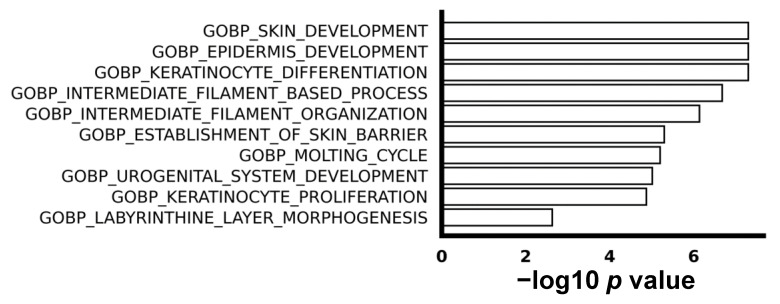
The top 10 enriched biological processes in gene oncology. GSEA shows top 10 pathways of biological processes.

**Figure 3 biomedicines-12-01101-f003:**
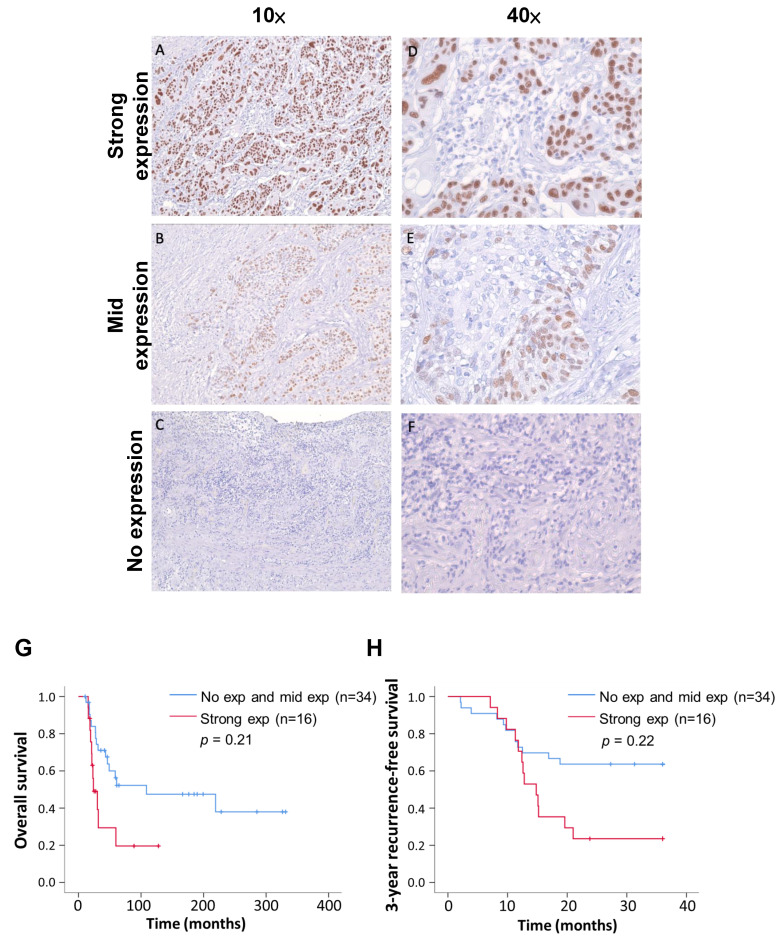
Immunohistochemistry indicates three TP63 expression levels; also shown are Kaplan–Meier overall survival and recurrence-free survival curves of 50 patients. (**A**,**D**) ESCC tissue showed strong TP63 expression at different magnification. (**B**,**E**) ESCC tissue showed medium TP63 expression at different magnification. (**C**,**F**) ESCC tissue with no TP63 expression. (**G**) Analysis of postoperative overall survival rates for different TP63 manifestations using the Kaplan–Meier estimator. (**H**) Comparative analysis of recurrence-free survival rates during three-year follow-up of different TP63 expressions.

**Figure 4 biomedicines-12-01101-f004:**
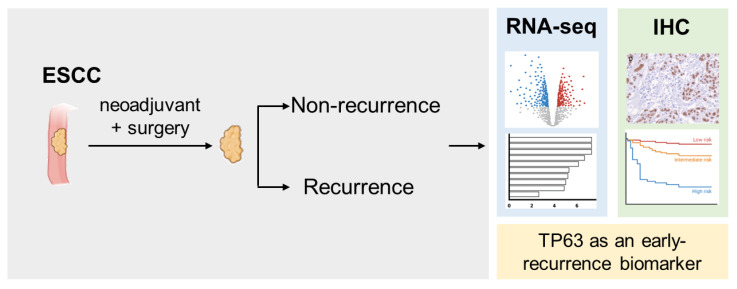
Graphic summary of the current study. This study found that TP63 expression is significantly increased in recurrent ESCC patients treated with nCCRT and surgery, as demonstrated by RNA-seq and IHC. Higher TP63 levels are associated with lower overall survival rates and 3-year recurrence-free survival, indicating TP63 as a potential biomarker for early recurrence prediction in ESCC.

**Table 1 biomedicines-12-01101-t001:** Characteristics of the enrolled ESCC patients.

Characteristics	Total	Recurrence	Non-Recurrence	*p*
No. of patients	N = 15	N = 8	N = 7	
Age (year, mean)	56.60 ± 6.68	56.25 ± 6.90	57.00 ± 7.34	0.84 ^a^
Gender				
Male	15 (100.0%)	8 (100.0%)	7 (100.0%)	
Female	0 (0%)	0 (0%)	0 (0%)	
Pathologic stage				
IA	3 (20.0%)	2 (25.0%)	1 (14.3%)	0.20 ^b^
IB	0 (0%)	0 (0%)	0 (0%)	
IIA	0 (0%)	0 (0%)	0 (0%)	
IIB	6 (40.0%)	1 (12.5%)	5 (71.4%)	
IIIA	4 (26.6%)	3 (37.5%)	1 (14.3%)	
IIIB	0 (0%)	0 (0%)	0 (0%)	
IIIC	1 (6.7%)	1 (12.5%)	0 (0%)	
IV	1 (6.7%)	1 (12.5%)	0 (0%)	

^a^ *t*-test analyzed the differences in age between patients with recurrent and non-recurrent ECSS patients. ^b^ Chi-square test analyzed the differences in each clinical stage between two groups.

**Table 2 biomedicines-12-01101-t002:** Top 20 up-regulated differentially expressed genes.

Gene	Base Mean	*p* Value	*p*adj	log2FC	Description
*SOST*	122.0422	7.55 × 10^−7^	4.01 × 10^−5^	9.820963	Sclerostin
*LINC02582*	87.1705	1.58 × 10^−5^	5.28 × 10^−4^	9.74846	Long Intergenic Non-Protein Coding RNA 2582
*AL033397.1*	85.11985	1.12 × 10^−5^	4.01 × 10^−4^	9.714346	Novel transcript
*DSG1*	632.4087	1.01 × 10^−12^	4.19 × 10^−10^	9.408987	Desmoglein
*FLG*	210.4592	4.43 × 10^−11^	1.01 × 10^−8^	9.261304	Filaggrin
*SLC47A2*	148.4492	5.27 × 10^−7^	2.95 × 10^−5^	9.163991	Solute Carrier Family 47 Member 2
*ZIC2*	60.1167	5.26 × 10^−5^	1.44 × 10^−3^	8.798748	Zic Family Member 2
*ALDH3A1*	7252.081	1.72 × 10^−23^	2.08 × 10^−19^	8.665461	Aldehyde Dehydrogenase 3 Family Member A1
*KRT1*	1000.965	4.73 × 10^−10^	7.32 × 10^−8^	8.440137	Keratin 1
*AC005336.1*	89.69816	1.15 × 10^−8^	1.15 × 10^−6^	8.028038	Inositol polyphosphate multikinase (IPMK) pseudogene
*S100A7 A*	173.8399	2.56 × 10^−8^	2.27 × 10^−6^	7.808537	S100 calcium binding protein A7
*AC245041.2*	116.7946	8.20 × 10^−18^	1.52 × 10^−14^	7.658909	Novel transcript
*TP63*	3486.077	7.02 × 10^−19^	1.88 × 10^−15^	7.549191	Tumor protein P63
*AKR1B10*	3507.269	8.13 × 10^−22^	4.90 × 10^−18^	7.488848	aldo-keto reductase family 1 member B10
*FAM83B*	263.4565	4.26 × 10^−17^	5.41 × 10^−14^	7.379748	Family With Sequence Similarity 83 Member B
*ALOX12P2*	132.6558	3.52 × 10^−15^	2.83 × 10^−12^	7.347922	arachidonate 12-lipoxygenase pseudogene 2
*KRT17*	52,363.25	1.96 × 10^−14^	1.31 × 10^−11^	7.333984	Keratin 17
*FOXE1*	268.5709	1.54 × 10^−13^	8.87 × 10^−11^	7.32836	Forkhead Box E1
*CCDC190*	203.4276	7.11 × 10^−13^	3.18 × 10^−10^	7.308247	Coiled-Coil Domain Containing 190
*CYP4F11*	1048.794	2.05 × 10^−11^	5.44 × 10^−9^	7.271066	Cytochrome P450 Family 4 Subfamily F Member 11

**Table 3 biomedicines-12-01101-t003:** The chi-square test results indicate a significant difference in the expression distribution of TP63 from IHC between recurrent and non-recurrent patients of ESCC.

	Total (N = 50)	Non-Recurrence (N = 25)	Recurrence (N = 25)	*p* Value ^a^
Strong expression	16	04 (25.0%)	12 (75.0%)	0.01 *
Medium expression	18	07 (38.9%)	11 (61.1%)	
No expression	16	14 (12.5%)	02 (87.5%)	

^a^ Chi-square test. * *p* < 0.05 was considered to be statistically significant.

## Data Availability

The results and the data analyzed during the current study are available from the corresponding author on reasonable request.

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
