# Peer review of "The Expression of TP63 as a Biomarker of Early Recurrence in Resected Esophageal Squamous Cell Carcinoma after Neoadjuvant Chemoradiotherapy"

_biomedicines, 2024, doi:10.3390/biomedicines12051101_

Round 1

Reviewer 1 Report

Comments and Suggestions for Authors

The work of the authors is interesting and relatively easy to follow. I have only minor points to raise:

- the methods should be expanded a little with more detail on the PC analysis and the genetic analysis

- better description of the population of the 50 case where the p63 were analyzed with IHC: why 25 and 25, wouldn't be better to have a series of consecutive patients without selection bias? This should also be acknowledged in the limitations of the study. Moreover, a description fo clinical-pathological characteristics of these patients should be added, even in a supplementary table: this would be useful for example whether a meta-analysis would be conducted on the role of p63 and your study could be included with all available data

- similarly, recurrence/survival rates in percentage should be added in the results

-regarding IHC, how many pathologists scored the cases? were there a measurement of their agreement? This should be added both in the methods and in the results, or acknowledged in the limitations of the study paragraph

Comments on the Quality of English Language

Minor English language polishing by a native speaker is advised.

Author Response

#Reviewer 1

Comments and Suggestions for Authors

The work of the authors is interesting and relatively easy to follow. I have only minor points to raise:

Q1.The methods should be expanded a little with more detail on the PC analysis and the genetic analysis

Answer:

Thanks for your reminder, detailed analysis methods have been added as follows:

“The principal components analysis (PCA) was also performed by the function “plotPCA” in R package DESeq2. The first two PCs were used to plot the results. For the gene set enrichment analysis (GSEA), the p-value metrics of all genes were converted to log form and the positive or negative sign was added based on the fold change estimate obtained from DESeq2. The genes were then pre-ranked by the trans-formed value before being input into GSEA software [PMID: 16199517, PMID: 12808457]. Gene sets of the gene ontology biological process (GO:BP) with sizes above 15 and under 500 were used in the analysis.” In Materials and Methods Section-2.2 RNA sequencing and gene expression analysis, page 3.

Q2.better description of the population of the 50 case where the p63 were analyzed with IHC: why 25 and 25, wouldn't be better to have a series of consecutive patients without selection bias? This should also be acknowledged in the limitations of the study.

Answer:

We document the limitations and reasons for the design of the study as follows:

“There were still several limitations in our study. First, it is a small-scale retrospective case-control study. The use of a case-control design may lead to selection bias, especially with a limited number of samples. This study aimed to confirm the correlation between recurrence and biomarkers, and we were concerned that a cohort study approach might result in too few recurrence cases to effectively address our research question. Therefore, we chose the case-control approach for numerical matching. In the future, we also hope to further validate our findings by expanding the sample size, evolving into a prospective study, or utilizing multi-center data to enhance the representativeness and generalizability of our results.” on page 4, line 283.

Q3.Moreover, a description for clinical-pathological characteristics of these patients should be added, even in a supplementary table: this would be useful for example whether a meta-analysis would be conducted on the role of p63 and your study could be included with all available data

Answer:

We have included the clinical pathological characteristics of 50 ESCC patients analyzed with IHC in the supplementary materials (Table S1), and added a narrative on Page 4, line 167.

Table S1. Characteristics of the enrolled 50 ESCC patients form IHC.

Characteristics

Total

Recurrence

Non-recurrence

p

No. of patients

N = 50

N = 25

N = 25

Age (year, mean)

54.52 ± 7.71

53.32 ± 7.81

55.72 ± 7.57

0.28a

Gender

Male

48 (98.0%)

23 (92.0%)

25 (100.0%)

Female

2 (04.0%)

2 (08.0%)

0 (00.0%)

Pathologic stagec

I

1 (02.0%)

0 (00.0%)

1 (100%)

0.237b

IA

7(14.0%)

3 (42.9%)

4 (57.1%)

IA2

1 (02.0%)

1 (100%)

0 (00.0%)

II

1 (02.0%)

1 (100%)

0 (00.0%)

IIA

3 (06.0%)

1 (33.3%)

2 (66.7%)

IIB

16 (32.0%)

4 (25.0%)

12 (75.0%)

IIIA

8 (16.0%)

6 (75.0%)

2 (25.0%)

IIIB

7 (14.0%)

5 (71.4%)

2 (28.6%)

IIIC

2 (04.0%)

2 (100%)

0 (00.0%)

IVA

2 (04.0%)

1 (50.0%)

1 (50.0%)

aT-test analyzed the differences of age between patients with recurrence and non-recurrence ECSS patients.

bChi-square test analyzed the differences in each clinical stage between two groups.

cTotal n = 48 due to missing data for 2 patients.

Q4.similarly, recurrence/survival rates in percentage should be added in the results.

Answer:

We added this sentence below: “These 25 patients with recurrence experienced recurrence within 25 to 235 days (average 135.2 ± 53.5 days). All patients had an average age of 54.52 ± 7.71 years, including 48 males and 2 females.” on Page 4, line 169.

Q5.regarding IHC, how many pathologists scored the cases? were there a measurement of their agreement? This should be added both in the methods and in the results, or acknowledged in the limitations of the study paragraph

Answer:

The method for determining IHC intensity has been added to the section of Materials and Methods-2.3 Immunohistochemistry on page 3.

“Two experienced pathologists from VGHTC independently evaluated the staining intensity of TP63 in tumor cells. Staining intensity is scored on a scale of 0-2, with 0 indicating no expression, 1 indicating mid expression, and 2 indicating strong expression.”

Reviewer 2 Report

Comments and Suggestions for Authors

The manuscript is generally well written and carries significant novelty in the field. In my opinion, the manuscript can be accepted for publication after considering the improvement below:

(1) Section 5. Conclusion - The statement in the manuscript is not a conclusion. Please remove this statement and add a conclusion of the study.

(2) Many references used are old references. Suggest to renew some references and use the most recent 5 years publication.

(3) Quality of the figures can be improved. Some figures eg. Figure 3 is very blur. Suggest to use a higher resolution figure. 

Comments on the Quality of English Language

Minor editing of English language required

Author Response

#Reviewer 2

Comments and Suggestions for Authors

The manuscript is generally well written and carries significant novelty in the field. In my opinion, the manuscript can be accepted for publication after considering the improvement below:

Q1. Section 5. Conclusion - The statement in the manuscript is not a conclusion. Please remove this statement and add a conclusion of the study.

Answer:

Thank you for your reminder, we did miss this section. We added the paragraph below: “This study explores the differences in gene expression between groups of patients with ESCC who received neoadjuvant concurrent chemoradiotherapy combined with surgery. Our findings show that the TP63 gene and protein expression levels are significantly higher in the recurrence group, accompanied by increased keratinizing and epidermis development activity. Higher expression of TP63 is associated with lower overall survival rates and a three-year recurrence-free survival. We hope that TP63 can serve as a biomarker for predicting early recurrence in ESCC and provide a new direction for future treatment strategies.” In Conclusions section, page 4.

Q2. Many references used are old references. Suggest to renew some references and use the most recent 5 years publication.

Answer:

Thank you for your suggestions, we have updated several newer references in the article.

Q3. Quality of the figures can be improved. Some figures eg. Figure 3 is very blur. Suggest to use a higher resolution figure.

Answer:

We modified the Figure 3 and increased the pixels to dpi 600.
